# Rapid Expansion of Virus-Specific CD4^+^ T Cell Types in the CNS of Susceptible Mice Infected with Theiler’s Virus

**DOI:** 10.3390/ijms21207719

**Published:** 2020-10-19

**Authors:** Hyun Seok Kang, Wanqiu Hou, Byung S. Kim

**Affiliations:** 1Department of Microbiology-Immunology, Northwestern University Feinberg School of Medicine, Chicago, IL 60611, USA; samuel44kang@gmail.com (H.S.K.); wanqiu.hou@gmail.com (W.H.); 2Michelson Medical Research Foundation, Monrovia, CA 91016, USA

**Keywords:** virus, demyelination, inflammation, Th cells, FoxP3^+^CD4^+^ T cells

## Abstract

The infection of susceptible mice with Theiler’s murine encephalomyelitis virus (TMEV) induces a T cell-mediated demyelinating disease. This system has been studied as a relevant infection model for multiple sclerosis (MS). Therefore, defining the type of T cell responses and their functions is critically important for understanding the relevant pathogenic mechanisms. In this study, we adoptively transferred naive VP2-specific TCR-Tg CD4^+^ T cells into syngeneic susceptible SJL mice and monitored the development of the disease and the activation and proliferation of CD4^+^ T cells during the early stages of viral infection. The preexisting VP2-specific naive CD4^+^ T cells promoted the pathogenesis of the disease in a dose-dependent manner. The transferred VP2-specific CD4^+^ T cells proliferated rapidly in the CNS starting at 2–3 dpi. High levels of FoxP3^+^CD4^+^ T cells were found in the CNS early in viral infection (3 dpi) and persisted throughout the infection. Activated VP2-specific FoxP3^+^CD4^+^ T cells inhibited the production of IFN-γ, but not IL-17, via the same VP2-specific CD4^+^ T cells without interfering in proliferation. Thus, the early presence of regulatory T cells in the CNS with viral infection may favor the induction of pathogenic Th17 cells over protective Th1 cells in susceptible mice, thereby establishing the pathogenesis of virus-induced demyelinating disease.

## 1. Introduction

The infection with Theiler’s murine encephalomyelitis virus (TMEV) results in viral persistence in the CNS of susceptible mice and induces demyelinating disease similar to multiple sclerosis [1,2]. Thus, this system has been extensively used as a relevant infection model for multiple sclerosis. The manifestation of demyelination induced after TMEV-inoculation is apparently immune mediated [3,4,5,6]. Both immune responses and viral persistence in the CNS play important roles in the pathogenesis of demyelination [7,8,9]. Several recent studies further defined the relationship between the levels of viral persistence and antiviral immunity in the pathogenesis of demyelinating disease [10,11]. The level of antiviral T cell immunity is more critical than the viral persistence level for the pathogenesis of demyelinating disease [10]. Furthermore, the type of CD4^+^ T cell responses preferentially contribute to the pathogenesis of viral demyelinating disease, i.e., IFN-γ producing Th1 cells protect against the pathogenesis [12], and IL-17 producing Th17 cells promote the pathogenesis by interfering with the apoptosis of virus-infected cells and the cytolysis of target cells by CD8^+^ T cells [13]. In addition, higher levels of FoxP3^+^CD4^+^T cells, which regulate the virus-specific Th1 and CD8^+^ T cells, were found in the CNS and periphery of TMEV-infected susceptible SJL mice compared to those in infected resistant B6 mice [14]. However, it is unknown whether such regulatory T cells inhibit the production of a specific cytokine activated with viral determinants or how pathogenic Th17 cells are amplified after TMEV infection.

Recent studies indicated that a higher level of initial viral replication in various antigen presenting cells, including dendritic cells, macrophages, microglia and astrocytes in susceptible SJL mice compared to resistant B6 mice determines the level and type of early T cell responses affecting the protection from or pathogenesis of viral demyelinating disease [12,15]. High levels of viral replication in these antigen presenting cells provide strong innate immunity signaling via TLRs [15,16,17] and MDA5 [18] for the production of various chemokines and cytokines, which, in turn, steer the differentiation of T cell types and modify their effector functions. In addition, activation of the NLRP3 inflammasome and the downstream PGE_2_ promote the pathogenesis of TMEV-induced demyelinating disease by enhancing the production of IL-17 [19,20]. Moreover, PGE_2_ inhibits the protective functions of Th1, CTLs, and NK cells [21]. Proinflammatory cytokines, such as IFN-α/β, IL-1β, and IL-6, produced following viral infection, promote Th17 over Th1 differentiation [13,22,23]. In addition, a high level of IL-6 promotes the expression of PD-1 and PD-L1, which inhibit the protective function of CD8^+^ T cells [24,25]. Indeed, viral infection of DCs and other antigen-presenting cells showed the preferential induction of pathogenic Th17 cells over protective Th1 cells, whereas UV-inactivate TMEV or viral peptides preferentially induced Th1 cells over Th17 cells [13,15]. In addition, the potential role of vitamin D in the pathogenesis of viral demyelinating disease would be intriguing because Vitamin D appears to play important roles in enhancing innate immune responses and steering the development of T cell subpopulations in MS [26]. Nevertheless, it has not yet been investigated how early virus-reactive cells are amplified and differentiated into functionally distinct CD4^+^ T cells in the host after TMEV infection. Furthermore, it also remains unknown whether these cells are compartmentally amplified in the CNS or infiltrated into the CNS following differentiation and amplification.

In this study, we adoptively transferred naive VP2-TCR-Tg CD4^+^ T cells into SJL mice to address the above questions. Our results demonstrate that preexisting naive VP2-specific CD4^+^ T cells promote the pathogenesis of TMEV-induced demyelinating disease at high initial viral loads, but VP2-primed VP2-specific CD4^+^ T cells protect against the pathogenesis. However, under a low viral load, the preexisting naive virus-specific CD4^+^ T cells are protective, indicating that a balance between the levels of initial viral loads and virus-reactive CD4^+^ T cells plays a critical role in the pathogenesis of TMEV-induced demyelinating disease. Our results also indicate that naive virus-reactive CD4^+^ T cells start to primarily replicate in the CNS and deep cervical lymph nodes rather than in the periphery as early as 2 d post infection. In the CNS of virus-infected susceptible SJL mice, higher levels of FoxP3^+^CD4^+^ T cells were accumulated, and a higher proportion of Th17 cells compared to Th1 cells were generated from naive virus-specific CD4^+^ T cells. VP2-specific FoxP3^+^CD4^+^ T cells activated in vitro showed the inhibition of the function of Th1 but not Th17 in an epitope-specific manner, suggesting that the early presence of FoxP3^+^CD4^+^ T cells favors the preferential generation of pathogenic Th17 cells in the CNS. Therefore, FoxP3^+^ regulatory CD4^+^ T cells together with Th17 cells may further inhibit the development and function of the protective Th1 cells. These results reveal the importance of the initial environment of naive CD4^+^ T cell differentiation and amplification in viral infection-induced chronic inflammatory diseases.

## 2. Results

### 2.1. Comparison of the Development of Demyelinating Disease and CD4^+^ T Cell Responses in SJL and VP2-TCR-Tg Mice after TMEV Infection

To directly compare the development of TMEV-induced demyelinating disease in SJL and VP2-TCR-Tg mice, we infected these mice with TMEV (2 × 10^6^ pfu/mouse), and the development of clinical symptoms was assessed over 60 days (Figure 1A). The VP2-TCR-Tg mice developed clinical signs at a significantly accelerated rate (*p* < 0.0001) compared to the control SJL mice. This result is consistent with the above experiments, indicating that the presence of elevated levels of naive CD4^+^ T cells specific for viral determinants promotes the pathogenesis of TMEV-induced demyelinating disease. To further determine the types of virus-specific CD4^+^ T cells differentiated after TMEV infection, we assessed the proportions of Th1 (IFN-γ) and Th17 (IL-17) cells in the CNS of the above mice upon restimulation with viral determinants at 8 d postinfection (Figure 1B). The VP2-TCR-Tg mice showed an elevated proportion of IFN-γ producing CD4^+^ T cells reactive to VP2_72-86_ compared to the control SJL mice. However, the overall proportions of Th1 cells reactive to viral determinants (VP2_72-86_ and 3D_20-38_) were similar (6% vs. 7.7%). In contrast, markedly elevated proportions of Th17 cells were observed in the VP2-TCR-Tg mice reactive to VP2 (0.6 vs. 6.4%), as well as 3D (1.1 vs. 4%) determinants, compared to the control SJL mice. The overall number of Th1 cells producing IFN-γ in the CNS of the VP2-TCR-Tg mice was lower (5.1 × 10^4^ vs. 9.2 × 10^4^ cells/CNS), although the number of VP2-reactive cells was higher compared to the control mice (Figure 1C). In contrast, the overall number of Th17 cells producing IL-17 in the CNS of VP2-TCR-Tg mice was greater than two-fold (7.9 × 10^4^ vs. 5.1 × 10^4^ cells/CNS) compared to that of the control mice, respectively. These results indicate that a high level of naive virus-specific CD4^+^ T cells, and perhaps other adjacent CD4^+^ cells, preferentially differentiated into the pathogenic Th17 cell type in the CNS environment upon TMEV infection.

### 2.2. Histopathologic Examinations of the SJL Mice and VP2-TCR-Tg Mice Infected with TMEV

Histopathological evaluations of the spinal cords of control SJL mice and VP2-TCR-Tg mice infected with TMEV at 65 dpi were compared (Figure 2). The demyelination levels were determined after LFB staining and the levels of axonal damage were monitored following Bielschowsky silver staining. The levels of inflammation and lymphocyte infiltration were evaluated after H&E staining. Lymphocyte infiltration, demyelination, and axonal loss were observed in the white matter and meninges of the spinal cords of both the control and VP2-TCR-Tg mice. However, the levels of demyelination and axon loss were more widely spread and severe in the white matter of the spinal cords in the VP2-TCR-Tg mice, compared to those of the control mice. The cellular infiltration levels appear to be similar in the white and gray matter between the control and VP2-TCR-Tg mice. These histopathological results are consistent with the clinical signs of the mouse groups (Figure 1).

### 2.3. Relationship among the Viral Dose, VP2-Specific CD4^+^ T Cells, and the Development of Demyelinating Disease

Previous studies have indicated that the level of CD4^+^ T cells plays an important role in the protection and/or pathogenesis of TMEV-induced demyelinating disease [12,13]. However, the relationship between the levels of initial viral load, virus-specific CD4^+^ T cells, and the pathogenesis of demyelinating disease remains unknown. To determine the effects of varying viral doses and the preexisting virus-specific naive CD4^+^ T cells on the development of TMEV-induced demyelinating disease, we adoptively transferred purified normal or VP2-TCR-Tg CD4^+^ T cells (1 × 10^6^) into naive SJL mice and then infected them with varying doses (2 × 10^5^, 1 × 10^6^ and 5 × 10^6^ PFU) of TMEV (Figure 3A). SJL mice with elevated numbers of the initial virus-reactive CD4^+^ T cells developed more severe and rapid clinical symptoms after infection with high viral doses (1 × 10^6^ and 5 × 10^6^ PFU). In contrast, under low viral infection (2 × 10^5^ PFU), the high initial virus-specific CD4^+^ T cells played a protective role in the development of clinical signs compared to the mice that received normal SJL CD4^+^ T cells. Thus, the initial viral dose and the frequency of virus-specific CD4^+^ T cells appear to be critically important in the development of clinical demyelinating disease.

We further investigated the effects of the presence of high levels of activated vs. naive CD4^+^ T cells specific for a viral determinant at the time of viral infection on the development of demyelinating disease (Figure 3B). One group of SJL mice (*n* = 10) received VP2-TCR-Tg CD4^+^ T cells primed with VP2_72-86_ in vitro for 24 h. The second group of SJL mice (*n* = 10) received unprimed VP2-TCR-Tg CD4^+^ T cells, and the third group received normal SJL CD4^+^ T cells as the control. The results indicate that the presence of activated virus-specific CD4^+^ T cells protects the mice from the development of TMEV-induced demyelinating disease, whereas naive virus-specific CD4^+^ T cells promote the pathogenesis (Figure 3B). Therefore, the initial differentiation of virus-specific CD4^+^ T cells under the environment of active viral infection appears to play a critically important role in the pathogenesis of TMEV-induced demyelinating disease.

### 2.4. Proliferation of Adoptively Transferred VP2-TCR-Tg CD4^+^ T Cells in Various Organs after Viral Infection

To further assess the activation patterns of naive CD4^+^ T cells specific for a viral determinant after TMEV infection, purified naive VP2-TCR-Tg CD4^+^ T cells were labeled with CFSE and then adoptively transferred into normal SJL mice. The pattern of proliferation of the CFSE-labeled VP2-TCR-Tg CD4^+^ T cells was compared to that of the control CD4^+^ T cells lacking reactivity to the viral determinant at 2, 3, 4, and 7 days postinfection (Figure 4). The proliferation of virus-reactive CD4^+^ T cells was detectable in the CNS as early as 2 dpi, and progressively increased at 3 and 4 dpi, whereas no detectable proliferation was observed in the recipients with the control CD4^+^ T cells (Figure 4A). By 7 dpi, the majority of the VP2-reactive CD4^+^ T cells proliferated for several generations compared to the low divisions of normal CD4^+^ T cells. We further compared the patterns of VP2-reactive CD4^+^ T cell proliferation in the CNS, the deep cervical lymph nodes (dcLN), and the spleens at 2, 3, 4, and 7 dpi (Figure 4B). At 2 dpi, the VP2-reactive CD4^+^ T cells began to proliferate in the CNS and then dcLN, with hardly any in the spleen. However, at 3 dpi, vigorous proliferations for several generations were seen in the CNS and dcLN, and proliferation was also detectable in the spleen. The rate of proliferation appeared to increase in the dcLN and spleen at 4 dpi, while the rate slowed down in the CNS. The proliferation reached its maximum (>10 divisions) by 7 dpi in all organs. Taken together, these results strongly suggest that virus-reactive CD4^+^ T cells initially differentiated and expanded in the CNS and dcLN where viral antigens are readily available.

### 2.5. Early Accumulation of FoxP3^+^CD4^+^ T Cells in the CNS after Viral Infection

Because FoxP3^+^CD4^+^ T cells are known to regulate virus-specific T cell responses [14], we also assessed the levels of FoxP3^+^CD4^+^ T cells in the CNS of TMEV-infected mice (Figure 5). First, we compared the levels of FoxP3^+^CD4^+^ T cells in the CNS and spleens between susceptible SJL and resistant B6 mice after TMEV infection at 5 dpi (Figure 5A). Susceptible SJL mice showed a higher level of FoxP3^+^CD4^+^ T cells in the CNS compared to that in the resistant B6 mice. However, resistant B6 showed no differences in the proportions of FoxP3^+^CD4^+^ T cells between the CNS and spleen. Conversely, a three-fold higher proportion of FoxP3^+^CD4^+^ T cells was found in the CNS of susceptible SJL mice compared to that in the spleen. As early as 3 dpi, FoxP3^+^CD4^+^ T cells were found in the CNS of both the TMEV-infected B6 and SJL mice (Figure 5B). Interestingly, the CNS of the SJL mice at 3 and 5 dpi displayed as much as two-fold higher proportions of FoxP3^+^CD4^+^ T cells with CD25^−^, suggesting that FoxP3^+^CD4^+^ T cells in the CNS of virus-infected SJL mice may represent undifferentiated regulatory T cells [27]. However, the proportions of CD25^hi^FoxP3^+^CD4^+^ T cells, which represent activated regulatory T cells, were similar between the B6 and SJL mice. When FoxP3^+^CD4^+^ T cells in the CNS of TMEV-infected VP2-TCR-Tg mice were compared with the B6 and SJL mice (Figure 5C), the proportions of FoxP3^+^CD4^+^ T cells in the CNS of the VP2-TCR-Tg mice were shown to be even higher and persisted in the CNS even at 8 dpi compared to those of the SJL mice. The differences among these mouse groups appear to reflect the relative frequencies of virus-specific FoxP3^+^CD4^+^ T cells. The overall numbers of FoxP3^+^CD4^+^ T cells rapidly and proportionally (10–20 fold) increased from 3 dpi to 8 dpi in the CNS of all different mouse groups (Figure 5D). Thus, the potential function of virus-specific FoxP3^+^CD4^+^ T cells accumulated in the CNS of susceptible mice early after TMEV infection may contribute to the pathogenesis of TMEV-induced demyelinating disease.

### 2.6. Suppression of IFN-γ but Not IL-17 Production by VP2-TCR-Tg FoxP3^+^CD4^+^ T Cells

To further understand the potential role of FoxP3^+^CD4^+^ T cells in the pathogenesis of TMEV-induced demyelinating disease, we generated activated VP2-TCR-Tg FoxP3^+^CD4^+^ T cells in vitro using a mixture of retinoic acid, TGF-β, IL-2, and a T cell stimulant (VP2_72-86_ or anti-CD3/CD28 antibodies), as previously described [28,29]. Under this condition, 40–60% of CD25^+^CD4^+^ T cells expressed FoxP3 (Figure 6A,B). TMEV-infected DCs induced ~7% and VP2_74-86_-loaded DCs induced 2% FoxP3^+^CD4^+^ T cells from naive VP2-TCR-Tg CD4^+^ T cells in the absence of the added cytokines (not shown). These results suggest that the cytokine environment of virus-infected cells determines the induction of FoxP3^+^CD4^+^ T cells from naive CD4^+^ T cells. We further assessed the inhibitory function of VP2-reactive TCR-Tg FoxP3^+^CD4^+^ T cells against the proliferative responses and their cytokine production of VP2-TCR-Tg CD4^+^ T cells in the presence of the specific epitope peptide, VP2_72-86_ (Figure 6C). This result clearly indicates that the activated VP2-TCR-Tg FoxP3^+^CD4^+^ T cells did not inhibit the proliferative responses of CD4^+^ T cells reactive to the identical epitope. Furthermore, only the production of IFN-γ (but not IL-17) by the VP2-specific CD4^+^ T cells in response to VP2_72-86_ was significantly inhibited in the presence of activated VP2-TCR-Tg FoxP3^+^CD4^+^ T cells in a dose-dependent manner (Figure 6C). However, the epitope-preferred suppression of CD4^+^ T cell proliferation was previously demonstrated with TCR-Tg FoxP3^+^CD4^+^ T cells from mice infected with mouse hepatitis virus [30]. Therefore, epitope-specific FoxP3^+^CD4^+^ T cells from TMEV-infected susceptible mice may have a regulatory function limited to the production of a certain cytokine by the same epitope-recognizing CD4^+^ T cells.

## 3. Discussion

Previous studies have indicated that antigen-presenting cells infected with live TMEV preferentially induce pathogenic Th17 responses, while cells treated with UV-inactivated TMEV or peptide preferentially induce protective Th1 responses [13,15]. Our results in this study indicate that the presence of a high level of initial virus-reactive CD4^+^ T cells exacerbates the development of TMEV-induced demyelinating disease (Figure 1). However, under a low viral infection, a high level of initial virus-specific CD4^+^ T cells protects mice from the development of demyelinating disease. These results suggest that the development of TMEV-induced demyelinating disease is strongly affected by the balance between the initial viral load and the frequency of virus-specific CD4^+^ T cells. The overall proportion of Th1 cells reactive to viral determinants was similar, but the proportion of Th17 cells was markedly elevated in the recipients of naive VP2-reactive CD4^+^ T cells compared to the recipients of normal CD4^+^ T cells (Figure 2). These results strongly suggest that naive CD4^+^ cells are preferentially differentiated into a pathogenic Th17 type under high viral loads. Therefore, the levels of initial viral load and virus-reactive CD4^+^ T cells play critical roles in their expansion and differentiation leading to the protection or pathogenesis of virus-induced demyelinating disease.

TMEV infection, like other viral infections, induces various innate immune responses via TLRs [16,17,31], MDA-5 [18], and PKR [32,33,34], leading to the production of many different chemokines and cytokines, which critically affect the pathogenesis of TMEV-induced demyelinating disease. Among the important cytokines, the effects of TNF-α [35], IL-1β [23], IL-6 [13,24,25], IFN-α/β [36,37], and prostaglandin E2 [19], which either alter the type of T cell responses or inhibit immune responses in the development of demyelinating disease, have been investigated. Excessive levels of IL-1β and IL-6 promote the generation of Th17 cells and inhibit the protective antiviral functions of CD8^+^ T cells. In addition, high levels of IFN-α/β and PGE2 inhibit the early generation of protective virus-specific Th1 cells. Therefore, high viral loads with elevated virus-reactive naive CD4^+^ T cells in the CNS may be preferentially differentiated into a pathogenic Th17 CD4^+^ T cell subtype.

Because TMEV is a neurotropic virus [38,39], and the cytokine environment plays a key role in the differentiation of T cells [40,41], the sites of the initial CD4^+^ T cell amplification and differentiation may be critically important in the pathogenesis of TMEV-induced demyelinating disease. Our results indicate that the initial proliferation of virus-reactive CD4^+^ T cells occurred in the CNS and dcLN rather than the periphery as early as 2 dpi and progressively increased at 3 and 4 dpi. These proliferations reached their maximum by 7 dpi in all organs (Figure 4). Therefore, virus-reactive CD4^+^ T cells appear to be initially activated and differentiated in the CNS and dcLN, where viral loads are high and the cytokine environment is favorable for the generation of the pathogenic Th17 subtype. High levels of FoxP3^+^CD4^+^ T cells are present in the CNS of virus-infected mice as early as 3 d after infection [14,30]. These results are consistent with the proportion of FoxP3^+^CD4^+^ T cells in the CNS of TMEV-infected SJL mice at 3–8 dpi (Figure 5). Therefore, the first cell type accumulated at the site of viral infection appears to be FoxP3^+^CD4^+^ T cells.

The early induction of FoxP3^+^CD4^+^ T cells may reflect the function of TLR2-mediated signals [42,43] resulting from TMEV infections [19]. When such FoxP3^+^CD4^+^ T cells were removed, the viral loads in the CNS and the development of clinical signs were significantly reduced in susceptible SJL mice [14] but not in resistant C57BL/6 mice [44]. These results suggest that the presence of a high level of FoxP3^+^CD4^+^ T cells promotes the pathogenesis of demyelinating disease. However, as much as a two-fold higher proportion of FoxP3^+^CD4^+^ T cells in the CNS of virus-infected SJL mice displayed CD25^lo^ (Figure 5), suggesting that FoxP3^+^CD4^+^ T cells in the CNS may undergo further activation to become functional regulatory T cells [27,45,46]. Notably, high levels of CD25^-^FoxP3^+^CD4^+^ T cells were also observed in chronic HBV-infected patients [47] and patients with systemic lupus [48]. Recently, it was documented that CD25^lo^FoxP3^+^CD4^+^ T cells may lose their capacity for FoxP3 expression and undergo transdifferentiation into pathogenic Th17 cells [49,50]. Therefore, it is conceivable that some or most of the CD25^lo^FoxP3^+^CD4^+^ T cells may be converted into pathogenic Th17 cells under an environment of abundant cytokines such as IL-6 and IL-1β in the CNS of TMEV-infected mice [13,23,24].

In many chronic viral infections, including TMEV, FoxP3^+^CD4^+^ T cells appear to contribute to the pathogenesis by inhibiting protective T cell functions and consequently promoting viral persistence [14,51]. In contrast, FoxP3^+^CD4^+^ T cells may be beneficial in controlling prolonged T cell responses in acute viral infections [52,53]. Because it is known that CD25^lo^FoxP3^+^CD4^+^ T cells may not display regulatory functions [27], we generated CD25^hi^FoxP3^+^CD4^+^ T cells from the spleen of VP2-TCR-Tg mice in the presence of critical cytokines and antigens, as previously described [28,29,46]. Under these experimental conditions, up to 60% of the CD4^+^ T cells displayed FoxP3 and CD25^hi^ (Figure 6). The VP2-specific FoxP3^+^CD4^+^ T cells activated in vitro inhibited the production of IFN-γ but not IL-17 via VP2-specific CD4^+^ T cells in an epitope-specific manner (Figure 6), suggesting the underlying mechanism for the preferential generation of Th17 cells that promote the pathogenesis of demyelination. Epitope-specific FoxP3^+^CD4^+^ T cells in MHV-infected mice showed more efficient inhibition of the same epitope-specific CD4^+^ T cells compared to bulk FoxP3^+^CD4^+^ T cells [30]. Therefore, a lack of inhibition of the proliferative response by epitope-specific Th cells (Figure 6) was unexpected. Previous studies utilized FoxP3^+^CD4^+^ T cells generated in vivo from virus-infected mice [14] or further stimulated in vitro with anti-CD3 and anti-CD28 antibodies instead of the epitope peptide [30]. In addition, the target Th cells of FoxP3^+^CD4^+^ T cells were from non-TCR-Tg mice. Therefore, the differences in FoxP3^+^CD4^+^ T cell stimulation and/or the target Th cell population may affect the inhibition of Th proliferation. Nevertheless, our results here indicate that VP2 epitope-specific FoxP3^+^CD4^+^ T cells preferentially inhibit the production of IFN-γ but not IL-17 via the same-epitope reactive Th cells (Figure 6). Because virus-reactive IFN-γ-producing Th1 cells are protective, but IL-17-producing Th17 cells are known to inhibit Th1 development and cytotoxic T cell functions [13,54], FoxP3^+^CD4^+^ T cells, together with Th17 cells, may promote the pathogenesis of TMEV-induced demyelinating disease.

Taken together, results in this study suggest that the balance between the initial viral load and virus-specific CD4^+^ T cells plays a critical role in the development of TMEV-induced demyelinating disease. In addition, the initial activation status of virus-reactive CD4^+^ T cells when encountering viral infection determines the outcome of the disease. A high initial viral load drives the rapid accumulation of virus-reactive FoxP3^+^CD4^+^ T cells in the CNS and subsequent directional Th differentiation favoring the development of highly pathogenic Th17 cells.

## 4. Materials and Methods

### 4.1. Mice

The genomic DNA of the TCRα and TCRβ chains that recognize the TMEV capsid protein peptide VP2_74-86_ in the conjunction of I-A^s^ was obtained from the CNS-derived 6.9 T cell clone [13,55]. These rearranged genomic TCR genes were subcloned into shuttle vectors (pTα and pTβTRIS) and microinjected into eggs from (B6XSJL)F1 mice at the transgenic core facility at Northwestern University. Transgenic Founders were screened by PCR and backcrossed over 20 times with SJL mice. We deposited this TCR-Tg strain (018030, SJL.Cg-Tg (TcraTcrbVP2)1Bkim/J) at the Jackson Laboratory for distribution. Foxp3-GFP transgenic mice (Rudensky laboratory (Seattle, WA, USA), which were further backcrossed to SJL mice for 10 generations, were obtained from Dr. Bill Karpus at Northwestern University. The mice were housed at the Animal Care Facility of Northwestern University. Experimental procedures approved by the Institutional Animal Care and Use Committee (#2011-1316, 1 April 2011; #2013-2133, 12 June 2013) were used.

### 4.2. Synthetic Peptides and Antibodies

All synthetic peptides purchased from GeneMed (GeneMed Synthesis Inc., South San Francisco, CA, USA) were used as described previously [56]. All antibodies used in this study were purchased from BD Pharmingen (San Diego, CA, USA).

### 4.3. Viruses and Cell Lines

The BeAn strain of TMEV was used in this study. The virus was propagated and tittered in BHK-21 cells. BHK-21 cells were grown and maintained in Dulbecco’s modified Eagle’s medium supplemented with 7.5% donor calf serum.

### 4.4. TMEV-Induced Demyelinating Disease

Mice were anesthetized with isoflurane and then intracerebrally infected with 30 μL of TMEV solution containing 2 × 10^5^–5 × 10^6^ pfu into the right cerebral hemisphere of 6–8-week-old mice. Clinical symptoms of the disease were assessed weekly using the following grading scale: grade 0 = no clinical signs; grade 1 = mild waddling gait or flaccid tail; grade 2 = severe waddling gait; grade 3 = moderate hind limb paresis; and grade 4 = severe hind limb paralysis.

### 4.5. Adoptive Transfer of Isolated CD4^+^ T Cells

CD4^+^ T cells from the naive TCR-Tg and control SJL mice were purified using a negative isolation kit (Miltenyi Biotec, Gladbach, Germany). The purified CD4^+^ TCR-Tg cells were then transferred intravenously (1 × 10^6^ CD4^+^ T cells/mouse) into naive SJL mice. These mice were intracerebrally infected with varying doses of TMEV (2 × 10^5^, 1 × 10^6^ or 5 × 10^6^ PFU). The development of clinical signs in the recipient groups was monitored for over 45 days post infection.

### 4.6. Histopathology

Mice were perfused via cardiac puncture with 50 mL of PBS at 65 days postinfection with TMEV. Spinal cords from the TCR-Tg and control SJL mice were fixed in 4% formalin in PBS for 24 h and embedded in OCT. The spinal cords were sectioned at 6 μm and stained with hematoxylin and eosin (H&E) for inflammatory infiltrates, Luxol Fast Blue (LFB) for axonal demyelination, and Bielschowsky silver staining for axon loss and damage. Four different sections of the lumbar region of the spinal cords per experimental group were examined using a Leica DMR fluorescent microscope. The images were captured using an AxioCam MRc camera with the AxioVision imaging software (Göttingen, Germany).

### 4.7. Isolation of CNS-Infiltrating Mononuclear Cells (MNC)

After mice were perfused with sterile Hank’s balanced salt solution (HBSS), their excised brains and spinal cords were homogenized using steel mesh. CNS-infiltrating MNCs were separated in the 1/3 bottom fraction of a continuous 100% Percoll (Pharmacia, Piscataway, NJ, USA) gradient following centrifugation for 30 min at 27,000× *g*, as previously described [57].

### 4.8. Intracellular Staining of Cytokine Production

CNS-infiltrating MNCs were cultured in 96-well round bottom plates in the presence of relevant or control peptides as described previously [22]. The surface T cell markers and intracellular cytokines in the cultured cells were stained using allophycocyanin-conjugated anti-CD8 (clone Ly2) or anti-CD4 (clone L3T4) antibodies and a PE-labeled rat monoclonal anti-IFN-γ (XMG1.2) antibody. Cells were analyzed on a Becton Dickinson FACS Calibur or a FACS Sort flow cytometer after live cells were gated based on their light scattering properties.

### 4.9. IFN-γ and IL-17 ELISA

ELISA kits for mouse IFN-γ and IL-17 were purchased from BD Biosciences (San Diego, CA, USA) and R&D Systems, Inc. (Minneapolis, MN, USA), respectively. To assess the cytokine levels in splenic culture supernatants, diluted samples were incubated for 2 h with plate-bound capture antibodies after blocking for 1 h, as instructed by the manufacturers. Cytokine expression levels were measured at 450 nm after treatment with HRP-conjugated detection antibodies in the presence of the HRP substrate TMB (BioFX Laboratories, Owings Mills, MD, USA).

### 4.10. T cell Proliferation Assay

The proliferative activity of epitope-specific CD4^+^ T-cells in the mixed cultures of FoxP3^+^ and CD4^+^ 69TCR-Tg cells was measured based on the [^3^H]thymidine incorporation levels following stimulation in the presence or absence of the epitope peptides for 3 days. Cells were harvested after a pulse for 18 h with 1 μCi [^3^H]thymidine-deoxyribose, and ^3^H incorporation was measured using TopCount. Data are expressed as the CPM ± SEM of triplicates.

### 4.11. T Cell Proliferation Assay Using CFSE

VP2-TCR-Tg CD4^+^ T cells were purified using a negative isolation kit (Miltenyi Biotec). The purified TCR-Tg CD4^+^ T cells were labeled with CFSE (10 μM) as described previously [25] and transferred intravenously (3 × 10^6^ VP2-TCR-Tg CD4^+^ T cells) into naive SJL mice. CFSE-labeled CD4^+^ T cells from naive SJL mice (3 × 10^6^) were transferred as the control. These mice were intracerebrally infected with 1 × 10^6^ pfu of TMEV. The intensities of CFSE-labeled CD4^+^ T cells in the CNS, deep cervical lymph nodes, and spleens were analyzed to determine the divisions of the transferred T cells at 2, 3, 4, and 7 dpi.

### 4.12. In Vitro Generation of FoxP3^+^ T Cells

Splenic CD4^+^ T cells from the VP2-TCR-Tg mice were cultured with IL-2 (PeproTech, 100U), TGF-β (PeproTech, 20 ng/mL), and 10 nM retinoic acid (Sigma-Aldrich, St. Louis, MO, USA) in the presence of either anti-CD3/CD28 antibodies or VP2_72-86_ peptide for 4 d. CD4^+^ T cells (100,000 cells/well in 24-flat-well plates) were placed in 1 mL of media. The level of FoxP3^+^CD4^+^ T cells was analyzed using flow cytometry after labeling the cells with APC-conjugated anti-CD4 and PE-conjugated anti-FoxP3 antibodies (BD Pharmingen, San Diego, CA, USA). For the functional analysis of FoxP3^+^CD4^+^ T cells, GFP-labeled VP2-TCR-Tg FoxP3^+^CD4^+^ T cells from the spleens of FoxP3/GFP-Tg SJL mice crossed to VP2-TCR-Tg SJL mice were used for activation. The activated FoxP3/GFP-Tg VP2-TCR-Tg CD4^+^ T cells were isolated using MoFlo (Dako Cytomation, Glostrup, Denmark).

### 4.13. Statistical Analysis

The significance of differences in the mean values was determined by Student’s *t* test. Group comparisons were performed using a one-way analysis of variance (ANOVA) with a Tukey–Kramer post hoc analysis using the InStat Program (GraphPAD software, San Diego, CA, USA). Data are shown as the mean ± SD of 2–3 independent experiments or triplicates of one representative experiment from at least three independent experiments.

## Figures and Tables

**Figure 1 ijms-21-07719-f001:**
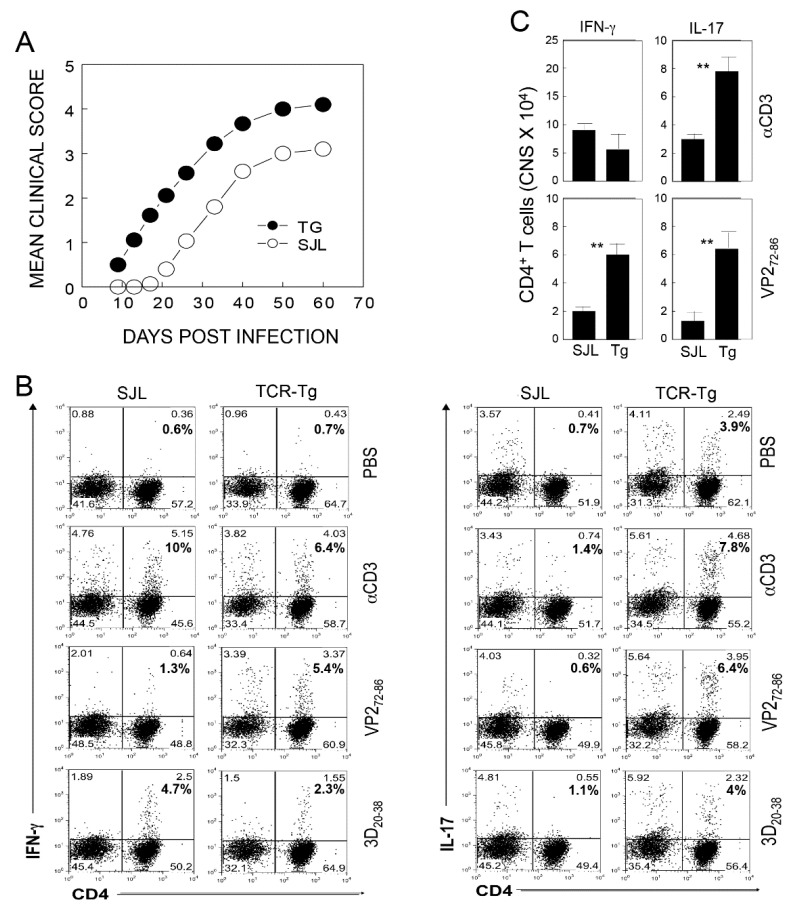
Effect of primed vs. naive CD4^+^ T cells specific for a viral determinant on the development of TMEV-induced demyelinating disease. (**A**) Control SJL and VP2-TCR-Tg mice were infected with TMEV (2 × 10^6^ pfu/mouse), and the development of clinical symptoms was compared between the groups over 60 days. The two-tailed *p* values between the groups were significant based on a paired *t* test of the mean clinical cores between days 9 and 60 postinfection: *p* < 0.0001 (*t* = 9.739 with 8 degrees of freedom) between the VP2-TCR-Tg group and the control SJL group. (**B**) Proportions of IFN-γ producing CD4^+^ T cells in the SJL and VP2-TCR-Tg mice. After 8 days of infection, CNS infiltrating cells were restimulated with PBS, anti-CD3/CD28, 2 µM VP2_72-86,_ or 3D_20-38_ peptides for 6 hr. The proportions of CD4^+^ T cells producing IFN-γ and IL-17 were determined using flow cytometry. (**C**) The numbers of CD4^+^ T cells producing IFN-γ and IL-17 in the CNS of TMEV-infected SJL and TCR-Tg mice at 8 dpi. ** *p* < 0.001.

**Figure 2 ijms-21-07719-f002:**
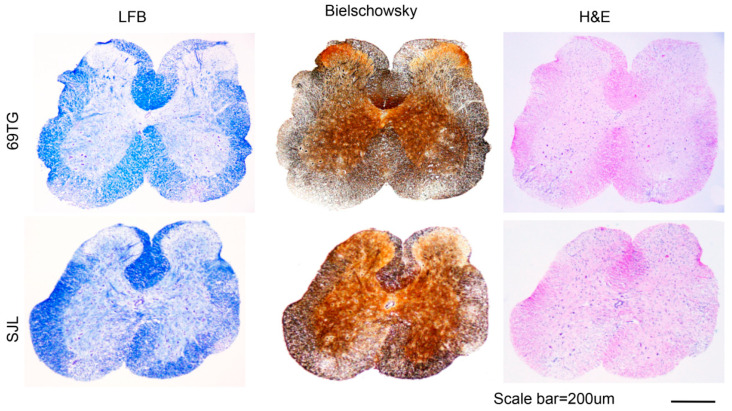
Histology of the spinal cords from TCR-Tg and control mice infected with TMEV. Four different sections of the spinal cords of TCR-Tg and control SJL mice at 65 dpi were stained with Luxol Fast Blue (LFB), Bielschowsky silver staining or H&E. A representative sample is shown. Original magnification of the black scale bar = 200 μm.

**Figure 3 ijms-21-07719-f003:**
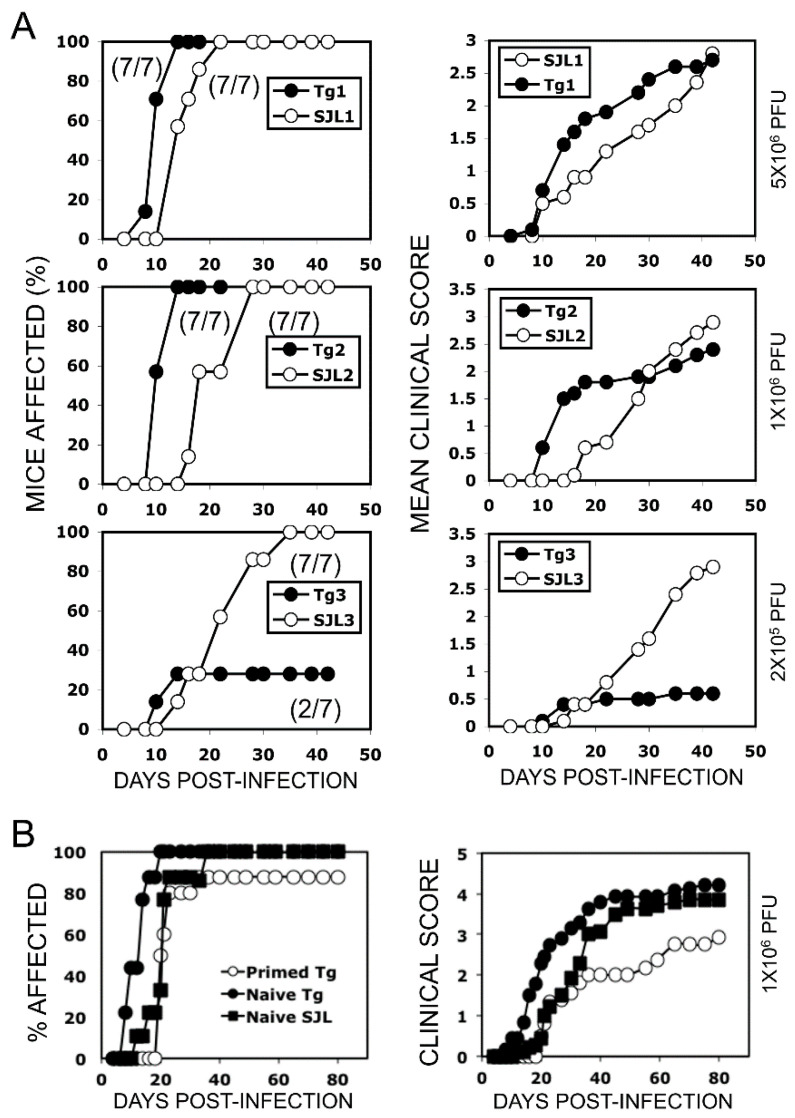
Effects of virus infection-dose and virus-reactive CD4^+^ T cells on the development of demyelinating disease. (**A**) Different viral doses (2 × 10^5^, 1 × 10^6^ and 5 × 10^6^ PFU) were used to infect SJL mice receiving VP2-TCR-Tg or control SJL CD4^+^ T cells to compare the development of clinical symptoms for 45 days. CD4^+^ T cells were purified using a negative selection column. Purified CD4^+^ T cells (1 × 10^6^/mouse) were injected through the tail-base vein of naive SJL mice. Recipient SJL mice (*n* = 7) were infected intra-cerebrally with the indicated PFUs; 5 × 10^6^ PFU (upper row), 1 × 10^6^ PFU (middle row), and 2 × 10^5^ PFU (bottom row). Clinical signs were scored two times per week for 50 days. The mean disease score (left column) and % of affected mice (right column) are shown. The two-tailed *p* values between the groups were significant based on the paired *t* test of the mean clinical cores, with *p* < 0.0001 (*t* = 17.080 with 7 degrees of freedom) in the upper row groups between days 10 and 35, *p* = 0.0219 (*t* = 2.309 with 6 degrees of freedom) in the middle row groups between days 10 and 30, and *p* = 0.0065 (*t* = 4.428 with 5 degrees of freedom) in the bottom groups between days 20 and 42. (**B**) Purified CD4^+^ T cells (2 × 10^6^ cells/mouse) stimulated in vitro with PBS or VP2_74-86_ for 6 h were transferred into naive SJL mice and then infected with TMEV (1 × 10^6^ pfu/mouse). The developments of clinical symptoms were compared among these experimental groups (*n* = 10) over 80 days postinfection. The two-tailed *p* values between the groups were significant based on a paired *t* test of the mean clinical cores between days 8 and 40 postinfection: *p* < 0.0001 (*t* = 8.534 with 13 degrees of freedom) between the group receiving naive VP2-TCR-Tg CD4^+^ T cells and the group receiving VP2-primed VP2-TCR-Tg CD4^+^ T cells, *p* < 0.0159 (*t* = 2.771 with 13 degrees of freedom) between the group with VP2-primed Tg CD4^+^ T cells and the group with naive SJL CD4^+^ T cells, and *p* < 0.0001 (*t* = 7.389 with 13 degrees of freedom) between the group with naive VP2-TCR-Tg CD4^+^ T cells and the group with naive SJL CD4^+^ T cells.

**Figure 4 ijms-21-07719-f004:**
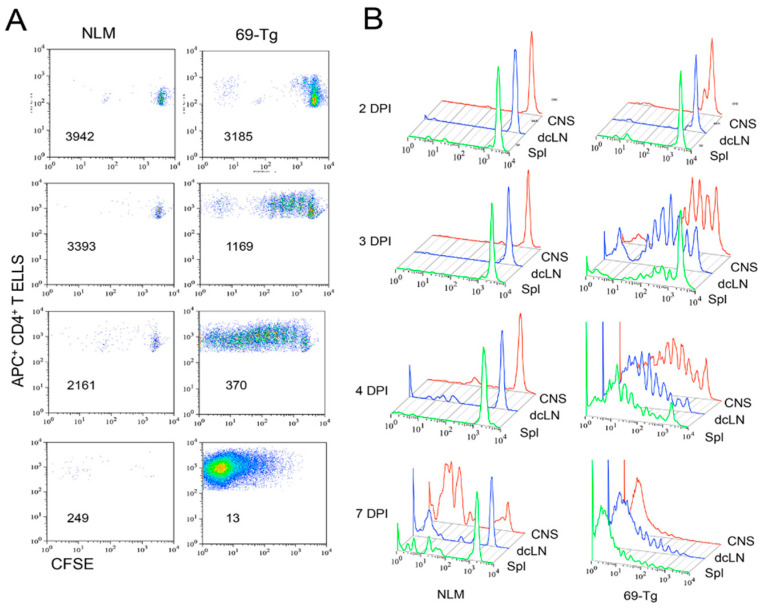
Comparison of the initial proliferative responses of TCR-Tg and control CD4^+^ T cells in the CNS and periphery after TMEV infection. Negatively isolated CD4^+^ T cells from naive TCR-Tg and control SJL mice were labeled with CFSE and transferred intravenously into normal SJL mice. The recipient mice were infected intracerebrally with 1 × 10^6^ PFU TMEV. Cells from the recipients from the CNS, deep cervical lymph nodes (dcLN), and spleens were analyzed at 2, 3, 4, and 7 dpi for their cellular divisions using flow cytometry after staining with APC-labeled anti-CD4^+^ T cells in conjunction with CSFE intensity. (**A**) The division patterns of CFSE^+^CD4^+^ T cells in the spinal cords of the mice receiving either control or TCR-Tg CD4^+^ T cells. The number in each flow cytogram represents the mean CFSE intensity of the CD4^+^ T cells. (**B**) Comparison of the division patterns of the transferred CD4^+^ T cells in the CNS, dcLN, and spleen.

**Figure 5 ijms-21-07719-f005:**
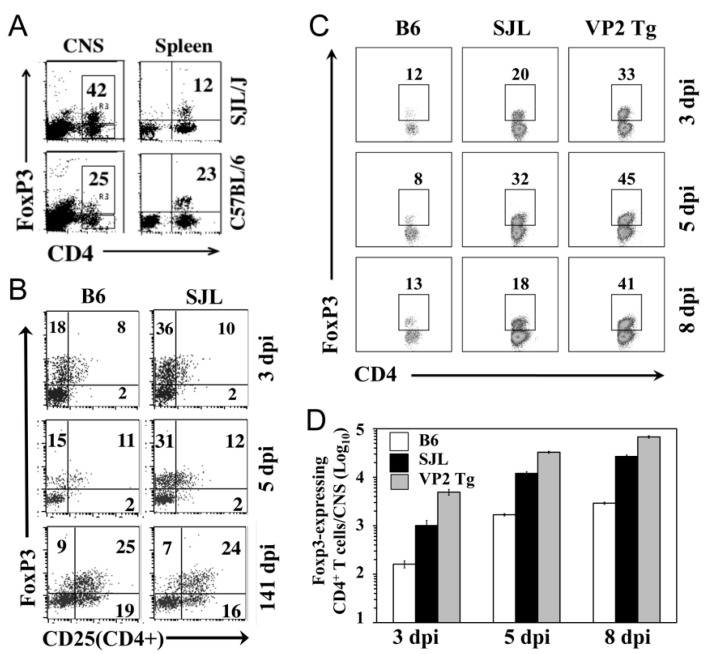
Comparison of FoxP3^+^CD4^+^ T cells in the CNS of C57BL/6, SJL, and TCR-Tg mice early after TMEV infection. (**A**) The proportions of FoxP3^+^CD4^+^ T cells in the CNS and spleens of TMEV-infected B6 and SJL mice at 5 d postinfection. (**B**) The expression of CD25 on FoxP3^+^CD4^+^ T cells in the CNS of TMEV-infected SJL and B6 mice at 3, 5, and 141 d postinfection. (**C**) The proportions of FoxP3^+^CD4^+^ T cells in the CNS of TMEV-infected B6, SJL, and TCR-Tg SJL mice at 3, 5, and 8 d postinfection. (**D**) The numbers of FoxP3^+^CD4^+^ T cells accumulated in the CNS of TMEV-infected B6, SJL, and TCR-Tg SJL mice at 3, 5, and 8 d postinfection.

**Figure 6 ijms-21-07719-f006:**
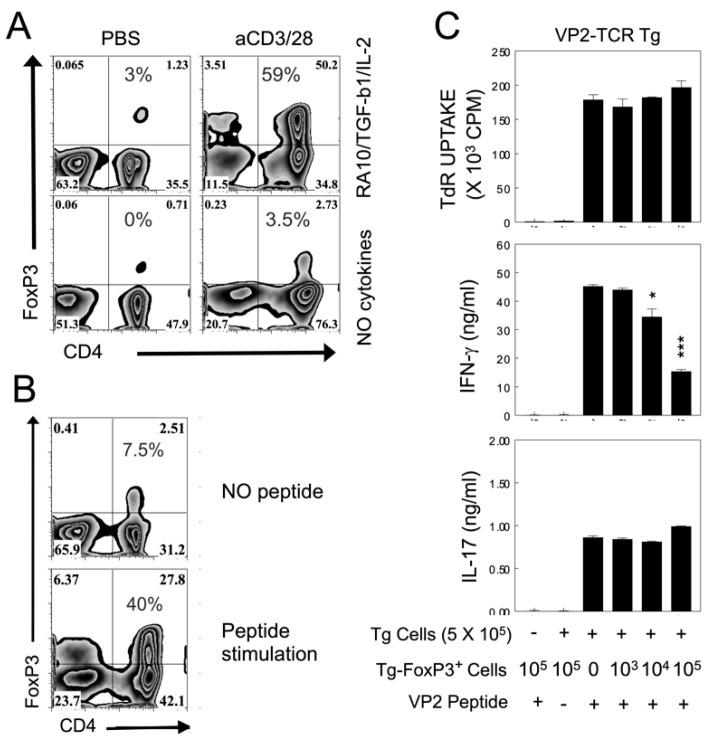
Role of VP2_72-86_ specific FoxP3^+^CD4^+^ T cells in the regulation of T cell cytokine production. (**A**) In vitro generation of FoxP3^+^CD4^+^ T cells in the presence or absence of T cell stimulants and/or added cytokines (10 nM retinoic acid, TGF-β, IL-2). Purified VP2 transgenic CD4^+^ T cells were activated with 1 µg/mL anti-CD28 and 10 µg/mL anti-CD3 plate-bound antibody in the presence of 20 ng/mL rhTGF-β1 100 U rhIL-2 for 4 d. (**B**) In vitro generation of VP2-specific FoxP3^+^CD4^+^ T cells in the presence or absence of VP2_72-86_ peptide in the presence of the above retinoic acid and cytokine mixture. (**C**) The function of VP2-TCR-Tg FoxP3^+^CD4^+^ T cells (10^3^–10^5^) activated in vitro was assessed with the inhibition of cytokine production of VP2-TCR Tg CD4^+^ T cells (5 × 10^5^) upon stimulation with the epitope peptide (VP2_72-86_). The VP2-TCR-Tg FoxP3^+^CD4^+^ T cells were purified from the spleens of FoxP3/GFP-Tg SJL mice crossed to VP2-TCR-Tg SJL mice for in vitro activation. The in vitro activated FoxP3^+^CD4^+^ T cells for 4 d were further sorted for GFP^+^ Tregs using MoFlo (Dako Cytomation, Glostrup, Denmark). * *p* < 0.05 and *** *p* < 0.0001. +, presence and −, absence of peptide.

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
