# Peer review of "Rapid Expansion of Virus-Specific CD4^+^ T Cell Types in the CNS of Susceptible Mice Infected with Theiler’s Virus"

_ijms, 2020, doi:10.3390/ijms21207719_

Round 1
Reviewer 1 Report
The manuscript by Kang et al. uses a TMEV model of demyelinating disease in mice to analyze the immune phenotypes that are involved in mediating disease severity. They present an elegant set of experiments demonstrating that a) pre-existing VP2-specific T cells increase viral pathogenesis at high viral doses, and reduce it at low viral doses, b) Th17 cells promote disease pathogenesis, and c) FoxP3+ Tregs do not in fact reduce disease pathogenesis, but instead increase it by limiting IFNg and inducing IL17 mediated response. Overall this is an important study showing potential pathogenicity of FoxP3+ Tregs.
Suggestions for improvement (no additional experiments requested, only data presentation):
1) Figure 1A: please provide error bars.
2) Figure 1B: please add quantification in addition to representative flow cytometry plots.
3) Figure 2: please add quantification in addition to images.
4) Figure 3: please provide error bars.
Author Response
Reviewer 1
1) Figure 1A: please provide error bars.
We did not show the error bars which are very much overlap and difficult to see the main differences. We have used paired parallel comparisons because two identical animal groups are parallelly compared at different time points. The detailed statistical analysis of the differences among the experimental groups in the development of demyelinating disease was described in the figure legend (L120-122).
2) Figure 1B: please add quantification in addition to representative flow cytometry plots.
At 8 dpi, the infiltration level in the CNS of TCR-Tg mice were significantly greater (1.5 fold) than that of SJL mice. Thus, the number of overall IFN-γ producing CD4+ cells was lower (5.1 x 104 cells/CNS vs. 9.2 x 104) and number of IL-17 producing CD4+ T cells was significantly elevated 3.2 x 104 vs. 7.9 x 104) in TCR-Tg mice compared to SJL mice. However, VP272-86 specific CD4+ T cell subpopulation among the CD4+ T cells remained higher in both IFN-γ and IL-17 producing CD4+ T cells. This information is presented in Figure 1C and now emphasized in the result section (L98-102).
3) Figure 2: please add quantification in addition to images.
It is very difficult to quantify the demyelinating lesions based on micrographs. Our intention to this figure is to show the similar in SJL and TCR-Tg mice, which are shown in Figure 2.
4) Figure 3: please provide error bars.
Please refer to the response to Figure 1A. We also provided the detailed statistical analyses of the differences among the experimental groups in the development of demyelinating disease. These are highlighted in the figure legend (L174-180).
Reviewer 2 Report
The study is well-conceived and well-presented. Rationale and results are intriguing, methods are clearly illustrated making the study reproducible.
However, the introduction could be improved, making it more complete. Indeed, while describing MS experimental model and T-helper lymphocytes subsets, a mention should be done about the role of Vitamin D, which bridges the Th subsets balance regulation to the pathogenesis of MS, as documented by several recent studies. Please, consider citing:
-PMID: 28834557
-PMID: 31142227
-PMID: 31284484
-PMID: 31330127
-PMID: 30588177
Author Response
Reviewer 2
The study is well-conceived and well-presented. Rationale and results are intriguing, methods are clearly illustrated making the study reproducible.
However, the introduction could be improved, making it more complete. Indeed, while describing MS experimental model and T-helper lymphocytes subsets, a mention should be done about the role of Vitamin D, which bridges the Th subsets balance regulation to the pathogenesis of MS, as documented by several recent studies. Please, consider citing:
-PMID: 28834557
-PMID: 31142227
-PMID: 31284484
-PMID: 31330127
-PMID: 30588177
We added a discussion on the potential role of Vitamin D in viral pathogenesis, including potential important roles in enhancing innate immune responses and steering T cell subpopulation development in the development of MS (L61-63).